Influence of droplet coverage on the electrochemical response of planar microelectrodes and potential solving strategies based on nesting concept

Yu Yue 1
Li Zhanming lizhanming@cjlu.edu.cn 1 2
1 Department of Biosystems Engineering, Zhejiang University , Hangzhou , China
2 Department of Food Science, College of Life Science, China Jiliang University , Hangzhou , China
Longo Marjorie
Electronic publication date: 2016 Aug 31
Publication date: 2016
Volume: 4
Electronic Location ID: e2400
Received 2016 May 10; Accepted 2016 Aug 3
Copyright: ©2016 Yu and Li
Copyright year: 2016
Copyright holder: Yu and Li
License: This is an open access article distributed under the terms of the Creative Commons Attribution License, which permits unrestricted use, distribution, reproduction and adaptation in any medium and for any purpose provided that it is properly attributed. For attribution, the original author(s), title, publication source (PeerJ) and either DOI or URL of the article must be cited.
License URL: https://creativecommons.org/licenses/by/4.0/

Keywords: Droplet coverage, Three dimensional (3D) printing, Signal change, Screen-printed interdigitated microelectrode (SPIM), Nest-like, Electrochemical response

Funding: China Jiliang University Start-Up Grant This work was supported by a China Jiliang University Start-Up Grant. The funders had no role in study design, data collection and analysis, decision to publish, or preparation of the manuscript.

==============================
Recently, biosensors have been widely used for the detection of bacteria, viruses and other toxins. Electrodes, as commonly used transducers, are a vital part of electrochemical biosensors. The coverage of the droplets can change significantly based on the hydrophobicity of the microelectrode surface materials. In the present research, screen-printed interdigitated microelectrodes (SPIMs), as one type of planar microelectrode, were applied to investigate the influence of droplet coverage on electrochemical response. Furthermore, three dimensional (3D) printing technology was employed to print smart devices with different diameters based on the nesting concept. Theoretical explanations were proposed to elucidate the influence of the droplet coverage on the electrochemical response. 3D-printed ring devices were used to incubate the SPIMs and the analytical performances of the SPIMs were tested. According to the results obtained, our device successfully improved the stability of the signal responses and eliminated irregular signal changes to a large extent. Our proposed method based on the nesting concept provides a promising method for the fabrication of stable electrochemical biosensors. We also introduced two types of electrode bases to improve the signal stability.

Introduction

Recently, biosensors have been widely used for the detection of bacteria, viruses and other toxins. Biosensors are analytical systems composed of a biological sensing element and a physical transducer. The transducer is designed for the conversion of biological information into a detectable signal, such as proton concentration, absorption or reflectance, light emission, mass changes, and so on (Hunt & Armani, 2010; Lepinay et al., 2014; Olaru et al., 2014; Thévenot et al., 2001; Van Dorst et al., 2010). Electrodes, as commonly used transducers, are vital part of electrochemical biosensors. They are designed to transform the recognition of a biological molecule into an easily quantifiable electrical signal (Gerard, Chaubey & Malhotra, 2002; Li et al., 2015b). Electrodes with small dimensions, commonly referred to as microelectrodes, can maximize the signal change, reduce the response time and work in a two-electrode system, thus benefiting the fabrication and performance of electrochemical biosensors (Huey et al., 2012; Park & Beskok, 2008). Microelectrodes exhibit more accurate electrochemical response to low concentrations of electro-active species in solution than large planar electrodes. Excellent flexibility and cycling stability also promise potential applications in lab-on-a-chip systems (Ch et al., 2006; Huang, O’Mahony & Compton, 2009; Ueno et al., 2005).

Electrochemical biosensors based on planar microelectrodes have been described in many literatures (Bernalte, Sánchez & Gil, 2011; Taleat, Khoshroo & Mazloum-Ardakani, 2014; Valentini et al., 2014; Wang, Ye & Ying, 2012; Zhu, Zhou & Gao, 1998). The sensitivity and the signal-to-noise ratio of planar microelectrodes can be influenced by many factors, such as the diameter of the electrode, surface coverage, electrode geometry and the electroactivity of analytes (Brett & Thiemann, 2002; Kostecki, Song & Kinoshita, 2000; Liu et al., 2014; Xu et al., 2004; Zhu, Zhou & Gao, 1998). The droplets exhibit different coverage on the microelectrode surface, depending on the hydrophobicity of the microelectrode surface which may trigger irregular signal change. The commonly-used incubation methods for planar microelectrodes include dropping and immersing coatings. The influence of droplet coverage which is important but less concerned, should be evaluated when dropping coating is used for incubation. Screen-printed interdigitated microelectrodes (SPIMs), as one type of planar microelectrode which may integrate the merits of screen printed microelectrodes and interdigitated microelectrodes, have been used to develop sensitive, rapid-responding, cost-effective biosensors (Li et al., 2015a). In this manuscript, we investigate the influence of droplet coverage on the electrochemical response of SPIMs.

Three-dimensional (3D) printed devices have captured much attention in many fields, including food safety and analysis (Sun et al., 2015; Xing, Zheng & Duan, 2015). As we know, many birds build nests to incubate their eggs and raise their young in a protective environment (Vilé et al., 2015). In order to further analyze the influence of the droplet coverage quantitatively, 3D printing technology was employed to print smart nest-like devices with different diameters to keep the immobilization and detection within the devices based on nesting concept. Both the theoretical analysis and our experimental results support our conclusion that the droplet coverage has significant influence on the electrochemical response of biosensors.

Depending on the varying hydrophobic properties of the materials used to immobilize the surface of the electrodes, the droplets can show gathering or dispersing properties (Costa, Pereira & Silva, 2015; Gryczan et al., 2015; Santoro et al., 2014). According to such principle, different materials are developed to remodel the surface of the SPIMs to improve the stability of signal responses. In the present research, ring devices around the detection area were designed to gather the droplets. Silica gel was used as a mode sample to validate our methodology. 3D-printed ring devices were used to incubate the SPIMs and the analytical performances of our device and method were also evaluated. Software can also be used to handle the signal stability by electrode base design.

Materials and Methods

Reagents and apparatus

E. coli O157:H7 (ATCC43889) was purchased from ATCC (Manassas, MD). Biotin-anti-E. coli antibodies was obtained from Meridian Life Science (Saco, ME) and dissolved in PBS solution (pH = 7.4). Bovine serum albumin (BSA), streptavidin (SA) and protein A were purchased from Sangon Biotech (Shanghai, China). MacConkey agar, brain heart infusion (BHI) culture medium were purchased from Becton, Dickinson and Company (Sparks, NV, USA). PBS solution containing 10 mM K3Fe(CN)6/K4Fe(CN)6 (Sangon Biotech., Shanghai, China) was used for electrochemical measurements. Ultrapure water (18.2 MΩ cm) was obtained from a Millipore Milli-Q purification system (Merck Millipore, Billerica, MA, USA).

The width of a finger and the gap between two fingers for SPIM (AIBIT Biotech Instrument, Jiangyin, China) are both 200 µm. One pair of gold electrodes and two welding plates were prepared on a ceramic base using screen-printed technology. Two electrodes were connected to the bonding pad. The electrode contained multiple conducting rings with different diameters and connected by conductive bands. Figure S1 showed the details of the electrodes.

Electrochemical impedance spectroscopy (EIS) and cyclic voltammetry (CV) were performed using ZAHNER electrochemical station (Kronach, Germany). Photosensitive resin was purchased from DSM SOMOS Crop. (Somos imagine 8000; Elgin, IL, USA). The devices with different diameters (2r = 6, 7, 8 and 9 millimeter) were printed by a 3D printer (Liantai 450, Shanghai, China) with 0.01 millimeter precision.

Experimental methods

Preparation of bacterial samples

E. coli O157:H7 was grown in BHI culture medium at 37 °C for 20 h to the stationary phase. The stationary-phase cultures were diluted to 107–101 cfu mL−1 in PBS (pH 7.4) and 100 µL of the diluted solutions were transferred to MacConkey agar plates and incubated at 37 °C for 24 h for enumeration of colonies. At the same time, the dilutions containing approximately 105 cfu mL−1 of bacteria cells were prepared for evaluation of the proposed devices.

Dropping and immersing coatings

When the planar microelectrodes were incubated, the commonly used incubation methods are dropping and immersing coatings for the incubation of planar microelectrodes. Usually, the volume of dropping coatings is 10 to 50 µL, and the volume of immersing coating is more than 1,000 µL in order to cover the whole detection area. Generally, the droplet coverage varies due to the discrepancy of the SPIMs surface. Figure 1 presents the droplet coverage of the electrode. The droplet coverage for the bare SPIM is significantly different compared to that for the modified SPIM (Fig. S2).

Furthermore, the droplet coverage may be changed during the modification process. Several factors that might have caused this include surface tension, gravity, interface hydrophobicity and the mobility of the molecules. In this experiment, different treatment groups were set up as comparisons. Treatment groups A and C were immersing coating and treatment groups PBS and B were dropping coating, respectively. We used 1,500 µL PBS solution containing 50 µL protein A for coating (0.5 mg mL−1) in treatment group A, 50 µL protein A (0.5 mg mL−1) in treatment group B, and 1,500 µL protein A (0.5 mg mL−1) in treatment group C. The signal change was detected with EIS or CV techniques.

Nesting concept for incubation

In order to investigate the difference caused by the droplet coverage, we designed and printed nest-like devices with different diameters. The devices (Fig. 2) possess an equivalent volume, but with different height and diameter (2r = 6, 7, 8 and 9 millimeter). The volume for these nest-like devices was equivalent, which ensured the same concentration and quantity of targets. The devices were incubated with BSA solution (2%, w/w) before use in order to avoid the non-specific absorption. Two devices (2r = 6 and 9 millimeter) were selected to perform the test to investigate the difference introduced by the multistep modification. The devices were used to incubate SPIMs with SA solution (50 µL, 0.5 mg mL−1) for 45 min, and then biotin-antibody was immobilized according to our previous research (Li et al., 2015a). After that, the bacteria solution was added into the devices to incubate the SPIMs and the difference caused by multistep modification was evaluated.

Figure 1 Detection area (droplets area) of the electrode.

Figure 2 The 3D printed nest-like devices (B) with different height (h) and diameter (2r = 6, 7, 8 and 9 millimeter) (C) and the photo of the incubation (D).

The bird nest picture (A) was cited from www.nipic.com/show/7011922.html.

3D-printing ring devices

In the present research, both resin and glass were used to investigate the dispersity of the target solution. A smart ring covered with silica gel was fabricated on the surface of the SPIM. We used the nest-like devices with different diameters to cover the detection area and then used silica gel to coat the devices. The caps were removed after the gel was completely dry in order to form a ring area. The ring can stay intact during the incubation and modification processes. In order to prove that the rings are intact, solutions with different colors were added into the devices after each incubation and washing processes (Fig. 3). The influence of NaOH solution was also investigated for the cleaning process.

Figure 3 Silica gel ring on the surface of the SPIMs.

Solution with different color were used to show the ring concept: (A) no solution, (B) solution with K+, (C) solution with Co3+, (D) solution with Cu2+, (E) ultrapure water.

On the basis of silica gel ring, we designed and printed ring devices with different diameters (Fig. 4). We selected two devices (2r = 6 and 9 millimeter) to investigate the performance. The surface of these devices were also blocked with BSA solution to avoid non-specific absorption. After that, these devices were fixed around of the detection area. The same amount of solution was added into the devices to incubate SPIMs for 45 min. Then, the non-specific absorption was washed with ultrapure water. The signal change was evaluated after the incubation.

Figure 4 (A) 3D printed ring devices with different diameters and (B) SPIM with ring device to form a nest-like device.

Figure 5 Immersing coating and dropping coating for the microelectrode modification.

Treatment groups A and C were immersing coating and treatment groups PBS and B were dropping coating. Treatment group A: 1,500 µL PBS solution containing 50 µL of protein A (0.5 mg mL−1); treatment group B: 50 µL of protein A (0.5 mg mL−1); treatment group C: 1,500 µL of protein A (0.5 mg mL−1).

Statistical analysis

Experiments were conducted in triplicate for each concentration level of protein A and other materials. In addition, bacteria cells were tested and the performance of the printed nest-like devices with different diameters were collected. Statistical analysis were conducted using SPSS 17.0. The biosensor responses were considered to show significant difference when P-value was less than 0.05 (95% confidence interval).

Results and Discussion

Dropping and immersing coatings

Four treatment groups were performed conducted and the application of PBS solution was used as a control (Fig. 5). Performances of treatment A and B were significantly different and the signal change of treatment B was more obvious than that of treatment A. Treatment C possessed more reactive molecules that can be captured on the surface of the devices in comparison with treatment B. These results indicates that the signal changes are different between immersing and dropping coatings. Since the two methods required different amount of materials, it is not persuasive to conclude that the difference is caused by droplet coverage.

Figure 6 Signal change of the printed nest-like devices with different diameters for protein immobilization.

Bare was a SPIM without printed devices.

Performance of nest-like devices

Considering the influence of the droplet coverage on the electrochemical response of SPIMs, a smarter design of devices is necessary. Nest-like devices with different diameters were used for incubation (Fig. 6). The results showed that there was significant difference between 9 millimeter and 6 millimeter devices (p > 0.05) and the difference were not significant for other devices. According to the difference in the droplet coverage of these two devices, it can be concluded that the signal response is influenced by droplet coverage.

In order to investigate the influence of the multistep modification, we selected 6 and 9 millimeters devices to perform the test for bacteria detection (Fig. 7). Treatment group A was prepared according to the signal change after SA modification. The results indicated that there was significant difference between the devices (p > 0.05). Treatment group B (Fig. 7) was prepared according to the signal change of the bacteria incubation after the multistep modification. The difference between the results collected from these two treatment groups did not disappear. Due to the large dimension of the bacteria cells, not every site was occupied by the bacteria cell. However, only effective absorption can introduce signal response. From the results, we concluded that the influence of droplet coverage was not disappeared by large dimension of cells. Therefore, it is clear that the droplet coverage is an influencing factor for electrochemical response.

Figure 7 3D printed nest-like devices (6 and 9 millimeters) for the test.

Treatment group A was the performance after SA modification. Treatment group B was the performance of the bacteria incubation, after the multi-step modification.

Figure 8 The CV performance of SPIMs with and without silica gel ring device.

Figure 9 The EIS (A) and CV (B) performance of the SPIMs with and without 3D printed ring devices (6 and 9 millimeters).

Figure 10 Two types of electrode bases designed by software.

(A) Interdigital microelectrode and (B) screen-printed electrode.

Performance of silica gel ring

Resin and glass were used to investigate the dispersity of the target solution. The solution coverage were different due to the different hydrophobicity of these two materials. However, the surface constructed by these two materials was completely (or partly) damaged when washing solution (NaOH, 1M) was used for SPIM regeneration. In order to deal with this situation, we firstly designed a ring was prepared on the surface of SPIM using silica gel. The results indicated that the signal was suppressed significantly compared to the treatment groups without the device, considering that more molecules were immobilized effectively (Fig. 8). Moreover, the washing solution was available for this device without damage, indicating that the material and device are both applicable.

Figure 11 The 3D printed base used to verify the application.

Performance of ring devices

We designed and printed smart ring devices based on the concept of the silica gel ring. The results in Fig. 9 show that the immobilization of the protein gave rise to the EIS and CV signal changes and the difference was significant between the two selected ring devices. The results showed the decreases of peak currents of the electrochemical probe by 27% (9 mm) and 68% (6 mm), and the increases of impedance signal by 25% and 45%, respectively, indicating the absorption was improved. Compared to the printed nest-like devices, less amount of materials is required for ring devices. Moreover, it is useful to avoid the leakage of solution during the process. It can reduce the evaporation as well, which is advantageous because avoid the irregular signal change can be avoided. Moreover, the devices can be recycled to decrease the detection cost because of no damage after an easy nondestructive cleaning process.

Base design using software

Signal stability can be influenced by the droplet area. Different bases can be designed to reduce the irregular signal. In this paper, we designed two bases using software (Fig. 10) that the droplets can be reserved within the scope of detection area. We used a 3D printer to print the base, verifying the application (Fig. 11).

Conclusion

In the present research, SPIMs was employed to evaluate the influence of droplet coverage on the electrochemical response. 3D printing technology was used to print fabricate mini-small smart devices with different diameters based on nesting concept. Nest-like devices with different diameters (2r = 6, 7, 8, 9 millimeter) were designed and printed to construct achieve different coverage to investigate the incubation performances and the results indicated that the influence of coverage on electrochemical response was significant. Moreover, ring devices on the surface of the SPIMs effectively improve the stability of the signal and also verify such influence. Detection cost can be greatly reduced by recycling the printed devices. All the devices improve the stability of the signal and successfully eliminate the irregular signal change. Our proposed design and concept shows great potential for application in the field of electrodes fabrication and stable electrochemical biosensors construction.

Supplemental Information

Figure S1 Details of the SPIM

Click here for additional data file.

Figure S2 Droplet coverage of the different SPIMs

Click here for additional data file.

Data S1 Raw Data

Click here for additional data file.

The authors thank the Biosensing & Biomodeling Lab at Zhejiang University for their support of the electrodes and electrochemical station. We gratefully acknowledge Muyang Lin and Xi Yu (College of Chemistry, National University of Singapore) for their help.

Additional Information and Declarations

Competing Interests

Author Contributions

Data Availability

The authors declare there are no competing interests.

Yue Yu conceived and designed the experiments, performed the experiments, analyzed the data, contributed reagents/materials/analysis tools, wrote the paper, prepared figures and/or tables, reviewed drafts of the paper.

Zhanming Li conceived and designed the experiments, performed the experiments, analyzed the data, contributed reagents/materials/analysis tools, wrote the paper, prepared figures and/or tables.

The following information was supplied regarding data availability:

The raw data has been supplied as a Supplemental File.

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
