# Peer review of "Influence of droplet coverage on the electrochemical response of planar microelectrodes and potential solving strategies based on nesting concept"

_PeerJ, doi:10.7717/peerj.2400_

## Round 0.1 · original submission · Major Revisions

Please address all of the comments of reviewer 1 regarding Experimental design and Validity of the findings. After we receive your revised manuscript, reviewer 1 may be asked to help determine if the comments were addressed sufficiently.

In addition, in my reading of the manuscript, I did not understand why bacteria was used in the nest configuration experiments but bacteria was not used in the ring configuration experiments. I expect this point to be clarified in the revised manuscript.

In addition, why was impedance spectroscopy used (shown) for the nested devices but cyclic voltametry was used (shown) for the ring devices? I expect this point to be clarified in the revised manuscript.

Reviewer 1 ·

Basic reporting

This manuscript reports a structural design of an electrochemical biosensor; the proposed simple but robust electrodes have been considered as one of the promising detection tools especially for small biomolecules. Although the geometric configuration of the electrodes are conventionally accepted and have been progressed with abundant of approaches as referred on this manuscript, there are still many rooms for in-depth researches or other viable approaches to be used for real applications.

Experimental design

They defined a method so called "screen-printed interdigitated microelectrodes (SPIMs)" and used in this paper. However, they did not explain the details of procedures and materials systems. On this, the minimal information, at least, should be supplied in the experimental section as listed below.

1. what materials that they used; materials system should be mentioned (e.g., carbon based materials, metallic surface, etc) so as to explain the details of surface morphologies of electrodes that they obtained. Because the printed materials directly can act with biomolecules onto the electrodes, this is important to see uniformity of the surface of the electrodes - this is critical because the CV data usually are affected by this.

2. the printing method; this is also influenced by the printing system. So, please provide the exact method using a schematic illustration or pictures of equipment. Also, experimental conditions to manipulate the electrodes should be mentioned such as printing conditions.

Validity of the findings

They supplied sufficient experimental data of fabricating the 3D printed "nests" to incubate the analytes effectively. The authors addressed the advantages of 3D printed ring adapted devices. However, in Figure 9, the signals looked similar although the sharp contrasts were generated. The readers might be confused what would be the main progress; I suggest the authors clarify the realtionship between the cycles of measurements and the changes of CV curves in Figure 9 with additional information.

---

## Round 0.2 · accepted · Accept

The authors have sufficiently addressed the comments with appropriate changes to the manuscript.